# Yield and combining ability of temperate ex-PVP introgressed and tropical maize under contrasting moisture regimes

Isaiah Aleri[1], Manje Gowda [1]*, Andrew Chavangi[1], Juan Burgueno[2], Yoseph Beyene[1]*

**1** International Maize and Wheat Improvement Centre (CIMMYT), Nairobi, Kenya, **2** International Maize and Wheat Improvement Centre (CIMMYT), Texcoco, Mexico

* y.beyene@cgiar.org (YB); m.gowda@cgiar.org (MG)

## Abstract

Low grain yield and recurrent drought remain major constraints to maize production in Sub-Saharan Africa (SSA), underscoring the urgent need for hybrids with improved resilience. The use of ex-PVP (expired Plant Variety Protection) temperate maize lines is important to improve yield potential and genetic diversity in tropical adapted lines. In this study, 15 temperate introgressed lines and 6 tropical lines were crossed with eight single cross testers to generate 168 testcross hybrids. All genotypes were evaluated in 8 sites, with 6 well-watered (WW) and 2 drought-stressed (DS) conditions in Kenya. Analysis showed significant genotypic and site effects for grain yield (GY) and other agronomic traits, with strong genotype × environment interactions under drought. General combining ability (GCA) and specific combining ability (SCA) were important, with additive variance dominating under both WW and DS. Temperate introgression (TI) improved yield performance in most tropical backgrounds, with hybrids from TI outperforming tropical hybrids by 1.20–2.04 t/ha under optimal conditions and yielding similarly under drought stress (TI, 1.88 t/ha; tropical, 1.89 t/ha). Hybrids showed up to 74% reduction in GY, 22% lower PH, 18% lower EH, and a 72% increase in ASI under DS compared to WW. Lines L2, L10, L16 and testers T1, T2, T3, T4, T7 showed consistently positive GCA for GY and shorter anthesis–silking interval, suggesting they can be used in breeding programs to develop high yielding drought tolerant hybrids suitable for sub-Saharan Africa. These findings emphasize the value of ex-PVP lines as source to enhance yield in tropical adapted germplasm. Both additive and non-additive gene action are important, combining pedigree methods with genomic selection for additive effects along with strategic heterosis use can accelerate the development of high-yielding, climate-resilient maize hybrids for stress-prone regions of SSA.

**Data availability statement:** The data is available in supplementary files.

**Funding:** The research was supported by the Bill and Melinda Gates Foundation (B&MGF), and the United States Agency for International Development (USAID) through the Stress Tolerant Maize for Africa (STMA, B&MGF Grant # OPP1134248) Project, AGGMW (Accelerating Genetic Gains in Maize and Wheat for Improved Livelihoods, B&MGF Investment ID INV-003439) project. The founders had no role in study design, data collection and analysis, decision to publish, or preparation of the manuscript.

**Competing interests:** The authors have declared that no competing interests exist.

## 1. Introduction

Maize (*Zea mays L.*) is the most widely grown cereal crop in Sub-Saharan Africa (SSA), serving as a staple food and a major source of income for millions of small-holder farmers. However, maize yields in SSA remain low, averaging only ~2.0 t ha$^{-1}$, compared to a global average of 5.8 t/ha [1]. This yield gap reflects a combination of factors including low-input production systems, slow adoption of improved varieties and agronomic practices, and the frequent occurrence of multiple biotic and abiotic stresses [2,3]. To sustain genetic gains, broadening genetic diversity is crucial. Introducing exotic maize germplasm into local breeding programs can expand the genetic base through new alleles. For example, the International Maize and Wheat Improvement Centre (CIMMYT) has introduced KS23−5 and KS23−6 as sources of Maize Lethal Necrosis (MLN) resistance [4] and developed several MLN tolerant lines from KS23 background.

Inbred lines of maize with expired Plant Variety Protection (ex-PVP) status represent an increasingly important pool of exotic germplasm. Under the International Convention for the Protection of New Plant Varieties, protection rights typically expire after 20 years [5,6]. In the United States, once protection expires, these lines become publicly available through the North Central Regional Plant Introduction Station (NCRPIS) at Ames, Iowa. Ex-PVP maize lines, originally developed by private breeding programs in temperate regions, have been widely evaluated for traits such as grain yield [7], nitrogen use efficiency [8], and stress tolerance. Their expired intellectual property status makes them accessible to both public and private breeding programs, providing a cost-effective means of accessing elite genetic material that has been extensively tested under temperate conditions. US Corn Belt germplasm contributed over 50% to modern maize hybrids in China, increasing yields [9], and sorghum introgression of short-stature and photoperiod-insensitive alleles improved temperate adaptation [10]. These cases demonstrate how exotic germplasm, when carefully introgressed, can strengthen breeding pipelines by introducing favorable alleles while broadening the genetic base.

Several studies have explored the potential of ex-PVP lines in hybrid development and pre-breeding. Studies by [11,12] showed some ex-PVP testcrosses underperforming, but others had desirable traits, suggesting their potential for pre-breeding. Cupertino-Rodrigues et al. [13], Ndoro [14], and Ndoro et al. [15] reported ex-PVP hybrids outperforming commercial checks, indicating they harbor beneficial alleles for yield and resilience. Temperate germplasm exhibit high yield potential and stability, hence making them superior compared with tropical germplasm [16]. The introgression of temperate germplasm could widen the genetic base of tropical germplasm. However, effective integration of maize germplasm in breeding requires accurate characterization of line performance and line relationships to other germplasm.

Line × tester analysis is a robust approach to evaluate the performance of diverse germplasm sources. The method estimates general combining ability (GCA), reflecting additive genetic effects, and specific combining ability (SCA), which captures non-additive gene action [17]. In maize, line × tester designs have been widely used to identify parental combinations with superior combining ability for yield and adaptive traits across contrasting environments [18–21]. To know which exotic germplasm had the highest potential and in which heterotic groups fit, the introduced inbred lines

along with the best adapted elite lines will be crossed with known testers and evaluated across the target environments to prioritize which exotic germplasm to be used in breeding prior to their effective utilization in a breeding program. Effective utilization of maize germplasm in breeding requires accurate characterization of line performance and line relationships to another germplasm using line by testers [22]. When developing breeding populations, maize breeders should choose parents that (i) show superior performance for the traits of interest, (ii) maximize within-population variance for the traits of interest, and (iii) preserve heterotic patterns for maximum heterosis in hybrid development.

Tropical elite lines developed in CIMMYT provide key adaptive and stress-tolerance alleles, whereas ex-PVP germplasm represents a rich source of high yield potential with well-characterized productivity-associated genomic regions. Because many yield-related haplotypes in ex-PVP lines are already defined, their introgression can be effectively tracked and validated in tropical backgrounds. Drought is a major constraint to maize production in sub-Saharan Africa and is projected to intensify under climate change [23], necessitating germplasm that combines productivity with resilience. In this study, testcross hybrids derived from temperate ex-PVP–introgressed and tropical inbred lines were evaluated using a line × tester design under well-watered (WW) and managed drought stress (DS) conditions. The objectives were (i) to assess hybrid performance for grain yield and agronomic traits across moisture regimes and (ii) to estimate general and specific combining ability for yield and secondary traits. By integrating and systematically evaluating ex-PVP germplasm in tropical breeding backgrounds, this work examines the potential of temperate introgression to broaden the genetic base and support development of high-yielding, stress-tolerant maize hybrids for SSA.

## 2. Materials and methods

### 2.1 Development of ex-PVP introgressed lines

To generate the ex-PVP introgressed lines, eight ex-PVP inbreds were crossed with 13 drought-tolerant tropical lines adapted to SSA. The tropical lines served as female parents, while the ex-PVP inbreds were used as pollen donors. Staggered planting was implemented to synchronize flowering between the parents. Resulting $F_1$ ears were harvested and planted in CIMMYT breeding nurseries during the 2008A season at Agua Fría and Tlaltizapán, Mexico. From each cross, $F_1$ plants were backcrossed to the recurrent tropical parent to obtain $BC_1$ seed, which was then advanced through ear-to-row sib-mating and selfing at CIMMYT-Zimbabwe research station. The $BC_1F_2$ generation was initiated with 250 families per cross. In subsequent generations, visual selection based on germination, stand establishment, plant architecture, ear placement, ear filling, kernel quality, and resistance to key foliar diseases [gray leaf spot (*Cercospora zeae-maydis*), northern leaf blight (*Exserohilum turcicum*), and maize streak virus (*Mastrevirus*, *Geminiviridae*)] were applied. Combining ability was not considered during the line development process. The selected $F_{2:3}$ plants were selfed to produce $F_{3:4}$ families, which were planted at Kiboko, Kenya, under high plant density (80,000 plants ha$^{-1}$). Plants with reduced lodging, lower ear placement, and desirable agronomic traits were advanced through successive selfing to produce $F_{4:5}$ and $F_{5:6}$ generations. A total of 324 advanced lines were topcrossed to elite CIMMYT testers and evaluated in preliminary trials. Based on grain yield, agronomic performance, and stability, 15 ex-PVP introgressed lines were selected for the current study.

Six tropical lines were included as parents alongside the ex-PVP introgressed lines (Table 1). Among the tropical lines, CKDHL0224 is a doubled haploid (DH) line derived from La Posta Seq C7-F71-1-2-1-2-B-B-B/ CML539//CML539 back-cross population. La Posta Seq C7 is a drought-tolerant population introduced to SSA from Mexico, while CML539 is an early maturity inbred line derived from CML312, a subtropical elite CIMMYT line. Further, the other line, CML537, is a tropically adapted derivative of CML312, while CML491 and CML495 are tropical late maturity inbred lines. These lines are extensively used as parents of commercial hybrids released across Eastern and Southern Africa.

### 2.2 Testers and hybrid formation

The 21 maize lines (15 ex-PVP introgressed and six tropical) were crossed to eight elite single-cross (SC) testers from the complementary heterotic group, following a balanced line × tester mating design (Table 1). In total, 168 hybrids were

**Table 1. List of ex-PVP temperate introgressed, tropical lines (L1-L21) and single cross tester (T1-T8) parents used to form TWC testcross hybrids.**

| Code | Group | Line/Tester Name | Pedigree, description and attributes |
|---|---|---|---|
| L1 | Tropical | CKDHL0224 | (La Posta Seq C7-F71-1-2-1-2-B-B-B/CML539/CML539)-DH-13-B |
| L2 | TI | CKLTI0028 | (CML539*/OFP9)-4-1-2-2-2-B-B |
| L3 | TI | CKLTI0036 | (CML539*/OFP27)-3-1-1-2-2-B-B |
| L4 | TI | CKLTI0041 | (CML539*/OFP14)-2-1-1-1-2-B-B |
| L5 | TI | CKLTI0045 | (CML539*/OFP14)-2-1-3-1-2-B-B |
| L6 | TI | CKLTI0147 | (CML495*/OFP9)-1-1-2-2-2-B-B |
| L7 | TI | CKLTI0174 | (CML495*/OFP14)-1-6-1-1-1-B-B |
| L8 | TI | CKLTI0182 | (CML495*/OFP14)-3-1-2-2-2-B-B |
| L9 | TI | CKLTI0188 | (CML495*/OFP14)-3-2-1-1-2-B-B |
| L10 | TI | CKLTI0200 | (CML495*/OFP14)-8-2-6-1-2-B-B |
| L11 | TI | CKLTI0234 | (CML491*/OFP106)-3-1-2-3-1-B-B |
| L12 | TI | CKLTI0238 | (CML491*/OFP106)-3-1-3-3-1-B-B |
| L13 | TI | CKLTI0272 | (CML537*/OFP106)-5-2-2-3-2-B-B |
| L14 | TI | CKLTI0344 | ([CML312/[TUXPSEQ]C1F2/P49-SR] …*/OFP106)-1-1-3-2-1-B-B |
| L15 | TI | CKLTI0348 | ([CML312/[TUXPSEQ]C1F2/P49-SR]…*/OFP106)-1-1-5-2-1-B-B |
| L16 | TI | CKLTI0368 | ([CML312/[TUXPSEQ]C1F2/P49-SR]…*/OFP106)-3-2-2-1-2-B-B |
| L17 | Tropical | CML312 | S89500-F2-2-2-1-1-B, Subtropical, intermediate maturity, with good GCA for drought tolerance, GLS and TLB |
| L18 | Tropical | CML491 | (6207QB/6207QA)-1−4-#-2–2-B, lowland late maturity, white GPM line |
| L19 | Tropical | CML495 | (PNVABCOSD/NPH-28–1)-F32-B-1-B-1–2-B, lowland late maturity |
| L20 | Tropical | CML537 | MAS(CML206/CML312)-23-2-1-1-B, Intermediate maturity, good for low N tolerance |
| L21 | Tropical | CML539 | MAS(MSR/CML312)-117-2-2-1-B, Early maturity, partially resistant to MSV |
| T1 | Tropical | CKDHL0089/CKDHL0295 | Single cross parent of released and nominated hybrids |
| T2 | Tropical | CKDHL0089/CKDHL0323 | Single cross parent of released and nominated hybrids |
| T3 | Tropical | CKDHL0089/CKDHL0333 | Single cross parent of released and nominated hybrids |
| T4 | Tropical | CKDHL0089/CML395 | Single cross parent of released and nominated hybrids |
| T5 | Tropical | CKDHL0159/CKDHL0295 | Single cross parent of released and nominated hybrids |
| T6 | Tropical | CML312/CML395 | Elite single cross testers |
| T7 | Tropical | CML395/CML444 | Elite single cross testers |
| T8 | Tropical | CML489/CML444 | Elite single cross testers |

TI – Temperate introgression; L – inbred line; T – single cross tester.

generated. The eight testers are commonly used in CIMMYT's eastern Africa breeding program, represent different genetic backgrounds, with some used as females in commercial hybrids. The 168 hybrids were organized into three trials, each containing 56 testcrosses and four commercial checks. This design allowed for robust evaluation across contrasting water regimes.

## 2.3 Field evaluation and data collection procedures

The hybrids were evaluated during the main rainy season of 2015 across eight sites in Kenya, six under WW conditions and at two sites under DS conditions. The WW sites included Kakamega (34°45′E, 0°16′N, 1585 masl), Kaguru (37°66′E, 0°09′S, 1450 masl), Kirinyaga (37°19′E, 0°34′S, 1282 masl), Embu (37°41′E, 0°97′S, 1510 masl), Kiboko (37°75′E, 2°15′S, 975 masl), and Mtwapa (39°44′E, 3°56′S, 21 masl). Managed DS trials were conducted at Kiboko and Mbeere

(37°43′E, 0°09′S, 1126 masl). Optimal (WW) trials were planted between April and October, whereas DS trials were conducted between May and October 2015. The environmental characterization of the growing season, including rainfall and temperature, across six WW and two DS experimental sites is presented in Supplementary S1 Table, confirming the distinct environmental conditions. At Kiboko and Mbeere, WW trials were planted earlier, while DS plantings were delayed to ensure a dry, rain-free period from flowering to harvest. All WW trials were grown under fertilized and irrigated or rainfed conditions, depending on location. At Kiboko and Mbeere, irrigation was applied during crop establishment and vegetative growth to maintain adequate soil moisture. For DS trials, irrigation was withdrawn two weeks prior to the expected flowering date to impose water stress during the flowering and grain-filling stages.

At each site, the experimental entries were arranged in a 10 × 6 alpha-lattice design with two replications. Each entry was sown in two 5-m-long rows, with spacing of 0.75 m between rows and 0.25 m within rows, resulting in a target plant population of 53,333 plants ha$^{-1}$. At planting, two seeds were placed per hill and thinned to one plant per hill three weeks after emergence. Standard agronomic practices, including fertilizer application and weed management, were applied uniformly across trials following CIMMYT protocols.

Data collection was conducted at each site for days to anthesis (AD, days), estimated from the day of planting up to when 50% of plants within a plot started to shed pollen. Days to silking (SD) were estimated from planting to when 50% of the plants had extruded silk. The anthesis silking interval (ASI) was calculated as the difference between SD and AD. Plant height (PH) was taken as the average height of 10 randomly selected plants in centimeters (cm) from the base of the plant to where tassel branching begins, while ear height (EH) was estimated as the average of plants from the same plants from the base of the plant to the node bearing the upper ear. Disease related traits were assessed as described by Badu-Apraku et al. [24], where severity to gray leaf spot (GLS) disease was assessed on an ordinal scale of 1–9. Ear rot (ER) was estimated as a percentage of the number of rotten ears to the total number of ears harvested [24]. At harvest, grain moisture [MOI, measured using a moisture meter on grain sampled from the center of five representative ears per plot] were recorded. Grain yield (GY) was calculated using the field weight of ears per plot, a shelling percentage of 80, and adjusted to a moisture content of 12.5%. All trait measurements were performed according to the procedures outlined in the drought phenotyping protocols by CIMMYT [25,26].

## 2.5 Analyses of data

Phenotypic data were analyzed across locations separately by management using the Restricted Maximum Likelihood (REML) method embedded in Multi-Environment Trial Analysis with R (META-R) software [27]. Analysis of variance (ANOVA) across sites was implemented in lme4 of R using REML according to a linear mixed model as follows:

$$Y_{ijkl} = \mu + G_i + E_j + R_k(E_j) + B_l[R_k(E_j)] + GE_{ij} + \varepsilon_{ijkl} \tag{1}$$

where, $Y_{ijdkm}$ is the response value, $\mu$ is the overall mean; $G_i$ is the effect of the $i^{th}$ genotype; $E_j$ is the effect of the $j^{th}$ location; $R_k(E_j)$ is the effect of $k^{th}$ replicate within $j^{th}$ location; $B_l[R_k(E_j)]$ is the effect of $l^{th}$ incomplete block within location and replicates; $GE_{ij}$ is the effect $i^{th}$ genotype interacting with $j^{th}$ location, and $\varepsilon_{ijkl}$ is the residual effects assumed to are independent and identically distributed, $\varepsilon \sim iidN(0, \varepsilon_{ijkl}^2$

## 2.6 Line-by-tester mating design analysis

Line-by-tester analysis was conducted across environments, but within management conditions, check entries were excluded from the analysis. The total variance among testcross hybrids was partitioned into GCA variances due to lines and testers, and SCA variance due to line × tester interactions [28,29]. Data analysis was performed using Analyses of Genetic Designs with R (AGD-R) version 4 [30]. A joint linear mixed model was fitted to the multi-environment trial data to estimate variance components, assess GCA and SCA, and obtain best linear unbiased predictions (BLUPs) of genotypic performance (Equation 2). The model was:

$$Y_{ijdkm} = \mu + E_d + R(E)_{kd} + B(ER)_{mdk} + L_i + T_j + LT_{ij} + LE_{id} + TE_{jd} + LTE_{ijd} + \varepsilon_{ijdkm} \qquad (2)$$

where, $Y_{ijdkm}$ is the response value, $\mu$ is the overall mean; $E_d$ is the effect of $d^{th}$ environment; $R(E)_{kd}$ is the effect of $k^{th}$ replication within $d^{th}$ environment; $B(ER)_{mdk}$ is the effect of $m^{th}$ incomplete block within $d^{th}$ environment within $k^{th}$ replication; $L_i$ is the GCA effect of the $i^{th}$ line; $T_j$ is the GCA effect of the $j^{th}$ tester; $LT_{ij}$ is the SCA effect of the $i^{th}$ line by $j^{th}$ tester; $LE_{id}$ is the GCA effect of $i^{th}$ line interacting with $d^{th}$ environment; $TE_{jd}$ is the GCA effect of $j^{th}$ tester interacting with $d^{th}$ environment; $LTE_{ijd}$ is the SCA effect of $i^{th}$ line by $j^{th}$ tester interacting with $d^{th}$ environment and $\varepsilon_{ijdkm}$ is the residual effects assumed to are independent and identically distributed, $\varepsilon \sim iidN(0, \varepsilon_{ijdkm}^2)$. In this model, environment and replication within environment were considered fixed effects, while lines, testers, their interactions (GCA and SCA), and blocks were treated as random effects. Variance components were estimated using the REML method [30]. Significance of variance components and model comparison were assessed using the Likelihood Ratio Test [31,32].

### 2.7 Estimation of general and specific combining ability effects

GCA and SCA effects were estimated according to the procedures of Singh and Chaudhary [33]. The statistical significance of GCA and SCA effects was assessed using *t*-tests. Each combining ability estimate was divided by its corresponding standard error to compute a test statistic, which was then compared with the critical values of the *t*-distribution at the 5% probability level, based on the appropriate error degrees of freedom.

To assess the relative importance of additive and non-additive genetic effects in determining hybrid performance, Baker's ratio [34] was calculated from the variance components associated with GCA and SCA. A ratio approaching unity (1) indicates that hybrid performance is predominantly controlled by additive gene action. Conversely, a lower ratio reflects a stronger role of non-additive effects, highlighting the importance of heterosis.

The relative contribution of lines, testers, and their interactions to the total genetic variance was estimated following the procedure of Singh and Chaudhary [33]. In this approach, the variance components associated with the GCA effects of lines and testers, and the SCA effects of line × tester combinations were compared to determine their proportional influence. The contribution of each source of genetic variation was expressed as a percentage of the total variance explained by lines, testers, and their interactions.

## 3. Results

### 3.1 Variance components and heritability

The evaluation of testcross hybrids developed from tropical and temperate introgressed lines demonstrated considerable phenotypic variation for GY and agronomic traits across environments (Fig 1). Under WW conditions, the average performance of hybrids consistently performed better (7.43 t/ha) than under DS conditions (1.89 t/ha), with notable reductions in yield and associated growth related traits (such as PH, 223.60–278.71 cm for WW and 166.67–244.55 cm for DS; EH, 100.98–154.68 cm under WW; 81.72–131.67 cm for DS) under stress, as validated by Student's *t*-test (Fig 2A). Temperate introgression (TI) generally improved GY (7.64 t/ha) in testcross hybrids compared to their original tropical counterparts' lines (6.94 t/ha; Fig 2A). Specifically, for CML495, TI derivatives yielded between 7.6 and 8.2 t/ha surpassing the tropical hybrid's 6.7 t/ha. The most notable increase was observed in CML539 TI derivatives, with TI_1 reaching 8.4 t/ha relative to 7.2 t/ha in the tropical line (Fig 2B). Similarly, TI derivatives of CML312 and CML491 outyielded tropical hybrids by 0.6–1.2 t/ha, with yields ranging from 7.3 to 7.6 t/ha. In contrast, CML537 showed only marginal differences, with hybrids ranging 7.2–7.4 t/ha for both tropical TI hybrids (Fig 2B).

Partitioning of variance across environments showed highly significant genotype and genotype × environment interaction (GEI) effects for GY and all measured traits (Table 2). Under WW conditions, genotypic variance exceeded GEI for most traits except MOI, ER, and GLS. Heritability (H²) estimates were high for GY (0.76) and flowering traits (0.83–0.91),

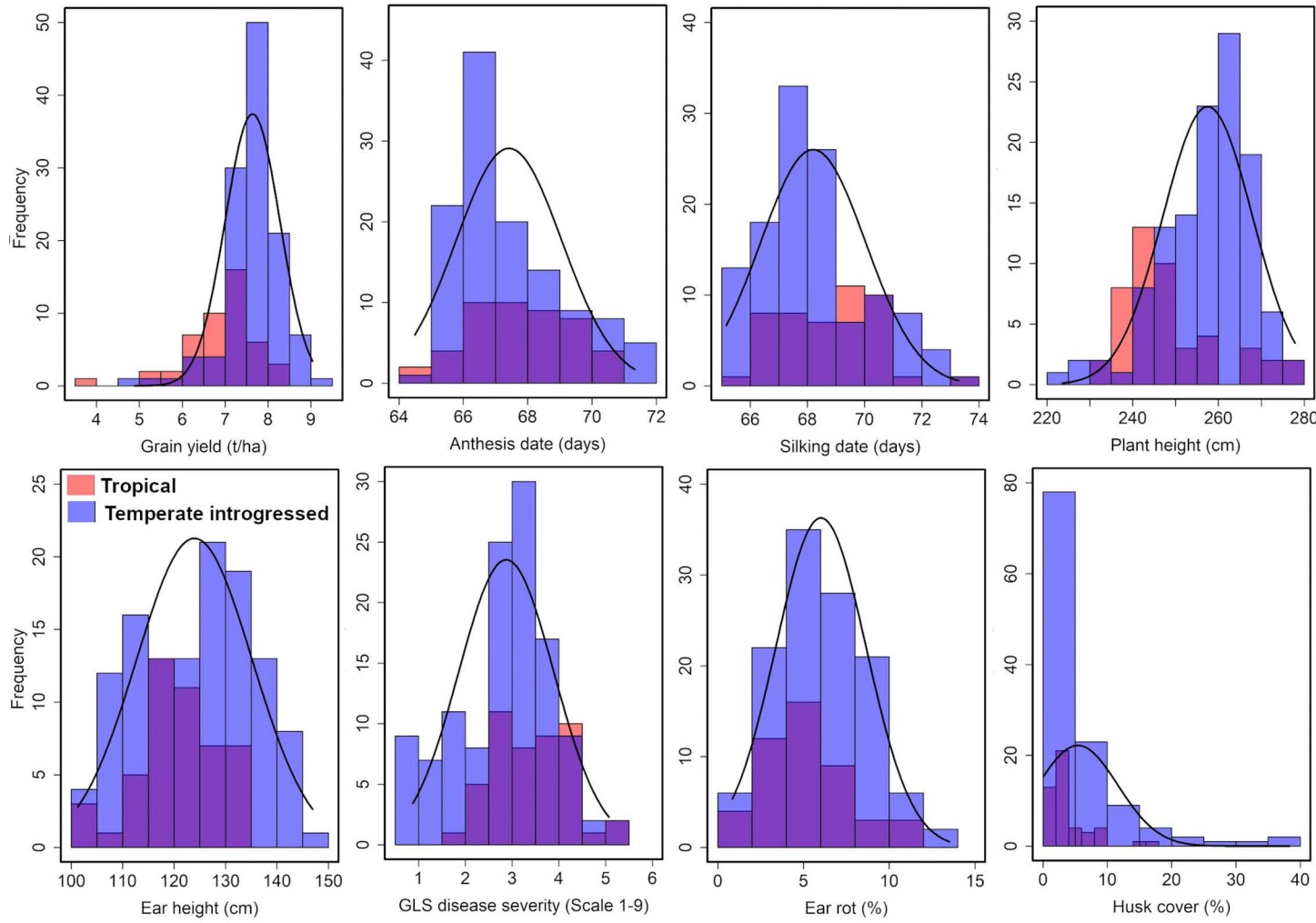

**Fig 1. Phenotypic distribution of testcross hybrids derived from tropical and temperate introgressed lines.**

but moderate for MOI, ER and GLS (0.41–0.45). Under DS, genotypic variance was often comparable to GEI variance for GY, AD, and EH, while GEI effects were larger for traits such as ASI, PH, and SD. As a result, heritability estimates under DS were lower, ranging from 0.18 for ASI to 0.54 for AD.

Under WW environments, mean GY was 7.43 t ha$^{-1}$, with relatively high heritability (0.76). In contrast, mean GY under DS dropped to 1.89 t ha$^{-1}$, with reduced heritability (0.33) and higher error variance, reflecting strong environmental influence. The flowering traits were more stable across environments. Under WW conditions, AD and SD showed high heritability (>0.89), but these values declined under DS conditions (0.54 for AD and 0.31 for SD). The ASI widened from 0.88 to 3.23 days under stress, with very low heritability (0.18). PH and EH also decreased markedly under DS, with mean PH dropping from 255 cm to 198 cm and EH from 123 cm to 100 cm (Table 2). Heritability for both traits was high in WW trials (>0.85) but fell below 0.40 under DS. For other agronomic traits, MOI showed moderate heritability under WW conditions (0.41) but was further reduced under DS (0.26). Disease-related traits (e.g., GLS) were assessed only under WW environments, where ER and GLS showed moderate heritability (0.40–0.45) but also significant GEI.

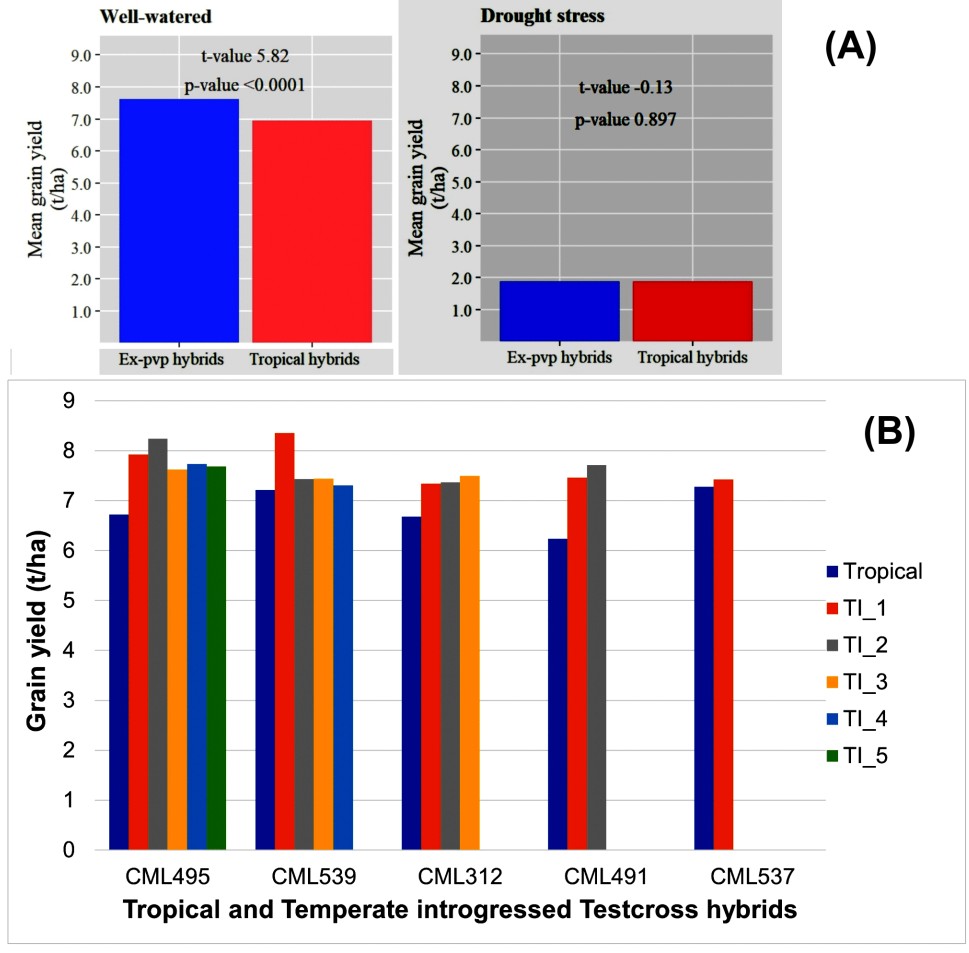

**Fig 2. Mean performance comparison of testcross hybrids from tropical and temperate-introgressed lines. (A)** Mean performance under well-watered (WW) and drought stress (DS) conditions compared using Student's *t*-test. **(B)** Grain yield comparison under WW between hybrids derived from original tropical lines and their corresponding temperate-introgressed versions. The x-axis denotes the original tropical parental line used for introgression; TI_1–TI_5 represent the five temperate-introgressed derivatives.

## 3.2 Performance of testcross hybrids

The mean performance of testcross hybrids derived from selected elite tropical lines and their TI lines and selected commercial hybrids as comparison checks, evaluated under WW and DS conditions are listed in supplementary tables (S2 and S3 Tables). Under WW conditions, GY of the top 10% hybrids was higher than the mean of checks (7.00 t ha), with values ranging between 8.20 and 9.04 t/ha (Table 3). The highest-yielding entry, T2/L7 (9.04 t/ha), substantially outperformed the mean performance of the checks (7.00 t/ha). Other strong performers included T3/L2 (8.88 t/ha), T5/L2 (8.79 t/ha), T2/L12 (8.77 t/ha), and T2/L2 (8.68 t/ha). The best hybrid (T2/L7) matched the best yielding commercial check for GY (9.04 vs 9.03 t ha$^{-1}$) but exhibited superior reproductive and architectural traits, including shorter ASI (−0.59 vs 1.38) and lower EH (116.7 vs 130.9 cm), indicating improved flowering synchrony and assimilate partitioning. On average, the top 10% hybrids yielded 8.51 t/ha, which was substantially higher than and the mean of checks (7.00 t/ha), highlighting their superior productivity under favorable growing conditions. Flowering traits in the top 10% hybrids under WW showed strong synchronization, with ASI averaging less than two days, and several hybrids such as T2/L7 and T1/L9 recording negative

Table 2. Estimation of total variance components for grain yield and other agronomic traits evaluated under well-watered and water stress locations in Kenya.

| Variance components | GY (t/ha) | AD (days) | SD (days) | ASI (days) | PH (cm) | EH (cm) | MOI (%) | ER (%) | GLS (1–5) |
|---|---|---|---|---|---|---|---|---|---|
| **Well-watered** | | | | | | | | | |
| Mean | 7.43 | 67.50 | 68.38 | 0.88 | 254.88 | 123.16 | 22.06 | 5.71 | 2.02 |
| $\sigma^2_G$ | 0.46** | 2.58** | 2.99** | 0.69** | 119.56** | 108.27** | 0.64* | 2.44** | 0.11* |
| $\sigma^2_{GxE}$ | 0.35** | 0.72** | 0.96** | 0.17** | 10.09* | 19.01* | 1.56** | 8.25** | 0.14** |
| $\sigma^2_e$ | 1.04 | 1.63 | 2.70 | 1.39 | 184.13 | 95.23 | 2.29 | 20.63 | 0.25 |
| $h^2$ | 0.76 | 0.91 | 0.89 | 0.83 | 0.85 | 0.89 | 0.41 | 0.40 | 0.45 |
| CV (%) | 13.76 | 1.89 | 2.40 | 134.14 | 5.32 | 7.91 | 6.86 | 79.59 | 24.82 |
| **Drought-stressed** | | | | | | | | | |
| Mean | 1.89 | 68.82 | 71.97 | 3.23 | 197.58 | 100.26 | 13.61 | – | – |
| $\sigma^2_G$ | 0.06* | 0.97** | 0.95** | 0.28* | 39.13** | 26.16** | 0.11* | – | – |
| $\sigma^2_{GxE}$ | 0.06** | 0.80** | 1.97** | 0.63** | 50.12** | 26.18** | 0.14** | – | – |
| $\sigma^2_e$ | 0.40 | 1.71 | 4.40 | 3.66 | 279.46 | 118.77 | 0.95 | – | – |
| $h^2$ | 0.33 | 0.54 | 0.31 | 0.18 | 0.29 | 0.38 | 0.26 | – | – |
| CV (%) | 33.3 | 1.90 | 2.91 | 59.12 | 8.46 | 10.87 | 7.16 | – | – |

*, **, Significant at P < 0.05; and < 0.01 probability levels, respectively. GY, grain yield; AD, days to 50% anthesis; SD, Days to 50% silking; ASI, anthesis-silking interval; PH, plant height; EH, Ear height; MOI, moisture content; ER, ear rot; and GLS, grey leaf spot disease severity.

ASI values. PH ranged from 230 to 270 cm, and EH from 108 to 138 cm, reflecting robust plant architecture. The incidence of ER was acceptable level with <10%, averaging between 4 and 7%, some hybrids (e.g., T1/L9 and T4/L2) displayed higher susceptibility compared to others.

In contrast, yields declined sharply under DS conditions, with the best hybrid, T5/L1, achieving only 3.43 t/ha. Other promising entries were T8/L20 (3.19 t/ha), T8/L2 (3.19 t/ha), T7/L2 (3.12 t/ha), and T7/L21 (3.12 t/ha). The best hybrid (T5/L1) yielded 3.43 t ha$^{-1}$, a 17% increase over the best check (2.93 t ha$^{-1}$), and associated with markedly reduced ASI (1.1 vs 5.8) and absence of ear rot. The average of the top 10% hybrids (2.87 t/ha) was markedly higher than the grand mean (1.89 t/ha) and nearly 40% greater than the mean of checks (2.09 t/ha), underscoring their superior adaptation to DS environments.

Under DS, flowering traits were more stressed, with ASI widening to about three to four days on average, compared to less than one day under WW. Both PH and EH were reduced, averaging around 201 cm and 103 cm, respectively, reflecting drought's impact on plant vigor. Interestingly, ER incidence was minimum under stress, with several hybrids, including T5/L1, T8/L20, and T3/L3, showing no visible ER, suggesting lower disease pressure in stressed environments.

### 3.3 Line by tester analysis

Under WW conditions, lines contributed most of the variance for flowering and plant height, while testers significantly influenced GY (Fig 3A). Line × tester interactions were significant across traits. Under DS conditions the contribution of lines was dominant for GY and most traits except PH and EH where testers and interaction of lines and testers contributed notably to the genotypic variance. Genetic variance analysis showed contrasting roles: additive and dominance effects mainly influenced GY under WW, whereas additive effects become more significant under DS (Fig 3B).

Line × tester analysis of variance confirmed highly significant differences among sites, genotypes, lines, testers, and their interactions for most traits under WW (Table 4). Under DS, site × genotype interactions remained significant, but the magnitude of genetic variance decreased, consistent with the reduced phenotypic expression of traits under stress.

**Table 3. Performance of top 10% (17) and lowest 5% (9) testcross hybrids for mean grain yield (GY, t/ha), days to anthesis (AD, days), days to silking (SD, day) anthesis silking interval (ASI, days), ear rot (ER), plant height (PH, cm), and ear height (EH, cm) under WW and DS conditions.**

| | Well-watered | | | | | | | | Drought stress | | | | | | |
|---|---|---|---|---|---|---|---|---|---|---|---|---|---|---|---|
| Hybrid | GY | AD | SD | ASI | PH | EH | ER | GLS | Hybrid | GY | AD | SD | ASI | PH | EH | ER |
| T2/L7 | 9.04 | 68.50 | 67.90 | −0.59 | 269.54 | 116.67 | 6.48 | 2.46 | T5/L1 | 3.43 | 69.01 | 70.00 | 1.10 | 198.78 | 113.63 | 0.00 |
| T3/L2 | 8.88 | 66.49 | 67.69 | 1.19 | 263.31 | 126.87 | 5.98 | 2.54 | T8/L20 | 3.19 | 68.23 | 70.25 | 2.02 | 192.07 | 102.35 | 0.00 |
| T5/L2 | 8.79 | 66.16 | 67.07 | 0.88 | 259.21 | 130.91 | 5.51 | 3.46 | T8/L2 | 3.19 | 69.96 | 71.95 | 4.09 | 214.54 | 103.78 | 0.83 |
| T2/L12 | 8.77 | 69.88 | 70.18 | 0.32 | 270.25 | 136.01 | 4.65 | 1.99 | T7/L2 | 3.12 | 68.53 | 72.17 | 4.09 | 197.07 | 111.77 | 4.02 |
| T2/L2 | 8.68 | 66.66 | 68.68 | 2.02 | 263.73 | 127.33 | 3.77 | 3.07 | T7/L21 | 3.12 | 67.26 | 71.75 | 4.47 | 192.09 | 98.98 | 10.93 |
| L7/L2 | 8.65 | 65.94 | 68.15 | 2.19 | 267.12 | 138.11 | 7.87 | 3.44 | T7/L9 | 2.95 | 68.34 | 69.83 | 1.19 | 198.08 | 101.38 | 6.00 |
| T8/L2 | 8.57 | 66.11 | 67.36 | 1.29 | 253.15 | 124.28 | 4.59 | 3.92 | T6/L13 | 2.91 | 67.49 | 72.00 | 4.28 | 191.76 | 91.66 | 15.29 |
| T1/L7 | 8.55 | 67.87 | 67.71 | −0.12 | 254.51 | 107.99 | 8.08 | 3.49 | T3/L3 | 2.89 | 67.06 | 70.00 | 3.19 | 209.48 | 101.31 | 0.00 |
| T7/L7 | 8.44 | 68.23 | 67.99 | −0.29 | 269.34 | 122.15 | 3.47 | 3.01 | T3L18 | 2.84 | 71.82 | 72.25 | 2.01 | 205.75 | 100.80 | 5.27 |
| T2/L6 | 8.42 | 67.96 | 68.74 | 0.80 | 268.50 | 119.38 | 6.91 | 3.60 | T2/L2 | 2.80 | 69.73 | 72.95 | 3.94 | 204.80 | 108.56 | 4.62 |
| T1/L2 | 8.37 | 66.40 | 68.50 | 2.16 | 256.82 | 127.58 | 6.23 | 4.42 | T5/L14 | 2.78 | 67.90 | 69.40 | 3.32 | 217.76 | 115.94 | 1.62 |
| T8/L7 | 8.35 | 68.37 | 68.10 | −0.31 | 250.69 | 113.57 | 4.62 | 3.43 | T7/L10 | 2.74 | . | 68.83 | 2.93 | 205.89 | 100.45 | 2.63 |
| T4/L2 | 8.27 | 66.51 | 68.77 | 2.26 | 266.05 | 129.87 | 9.07 | 2.46 | T2/L20 | 2.64 | 67.19 | 70.50 | 3.27 | 202.50 | 97.40 | 3.33 |
| T4/L7 | 8.23 | 68.46 | 68.38 | −0.08 | 267.79 | 116.46 | 5.63 | 3.88 | T4/L13 | 2.63 | 67.58 | 70.00 | 2.35 | 203.17 | 99.57 | 5.77 |
| T1/L9 | 8.22 | 66.02 | 65.35 | −0.63 | 230.04 | 108.79 | 8.21 | 2.93 | T7/L20 | 2.51 | 67.70 | 70.00 | 2.39 | 206.81 | 103.09 | 2.82 |
| T1/L4 | 8.21 | 69.87 | 70.43 | 0.54 | 266.06 | 137.54 | 2.06 | 3.91 | T8/L21 | 2.51 | 67.34 | 69.50 | 2.19 | 193.41 | 94.01 | 2.78 |
| T2/L4 | 8.20 | 69.44 | 70.17 | 0.74 | 268.00 | 130.77 | 2.11 | 3.89 | T4/L5 | 2.50 | 69.02 | 72.00 | 3.07 | 198.09 | 99.12 | 4.15 |
| T3/L18 | 6.17 | 69.78 | 70.71 | 0.93 | 242.08 | 119.35 | 4.70 | 2.47 | T4/L11 | 1.27 | 70.30 | 72.50 | 2.19 | 198.12 | 106.33 | 3.89 |
| T6/L15 | 5.79 | 66.69 | 67.98 | 1.31 | 273.04 | 127.83 | 6.94 | 1.60 | T8/L18 | 1.26 | 70.27 | 74.00 | 3.78 | 181.89 | 88.42 | 2.35 |
| T6/L20 | 5.73 | 66.11 | 68.79 | 2.64 | 244.67 | 104.50 | 6.13 | 3.48 | T3/L16 | 1.26 | 67.81 | 68.58 | 2.64 | 211.48 | 110.88 | 0.00 |
| T8/L18 | 5.70 | 70.80 | 70.75 | −0.10 | 238.99 | 121.82 | 3.22 | 4.44 | T4/L1 | 1.25 | 68.55 | 71.25 | 2.67 | 201.97 | 113.96 | 4.82 |
| T5/L18 | 5.47 | 70.07 | 70.57 | 0.50 | 242.64 | 120.90 | 5.54 | 4.00 | T2/L6 | 1.19 | 70.20 | 73.66 | 3.39 | 197.65 | 98.39 | 1.43 |
| T6/L21 | 5.44 | 65.33 | 69.19 | 3.88 | 230.32 | 101.91 | 2.53 | 4.07 | T5/L19 | 1.19 | 69.02 | 72.75 | 3.66 | 200.59 | 103.99 | 1.32 |
| T6/L5 | 5.24 | 69.61 | 71.95 | 2.37 | 223.60 | 110.89 | 2.05 | 2.63 | T1/L9 | 1.12 | 68.58 | 71.75 | 3.26 | 194.37 | 85.78 | 11.26 |
| T6/L4 | 4.89 | 70.22 | 73.29 | 3.12 | 254.87 | 117.82 | 6.50 | 4.51 | T4/L3 | 1.11 | 67.98 | 71.00 | 3.04 | 188.20 | 109.37 | 1.80 |
| T6/L17 | 3.71 | 70.34 | 73.37 | 3.06 | 235.70 | 100.98 | 5.35 | 3.00 | T1L1 | 1.08 | 68.96 | 70.75 | 1.62 | 188.07 | 96.62 | 5.56 |
| Min | 3.71 | 63.11 | 64.93 | −1.50 | 223.60 | 100.98 | 0.85 | 0.88 | Min | 1.08 | 65.99 | 67.42 | 0.85 | 166.67 | 81.72 | 0.00 |
| Max | 9.04 | 71.33 | 73.37 | 3.88 | 278.71 | 154.68 | 13.51 | 5.09 | Max | 3.43 | 72.42 | 82.10 | 12.04 | 244.55 | 131.67 | 15.29 |
| Mean of top 10% | 8.51 | 67.58 | 68.30 | 0.73 | 261.42 | 124.37 | 5.60 | 3.29 | Mean of top 10% | 2.87 | 68.38 | 70.79 | 2.94 | 201.88 | 102.58 | 4.12 |
| Mean of TI | 7.64 | 67.42 | 68.20 | 0.79 | 257.48 | 123.94 | 6.02 | 2.88 | Mean of TI | 1.88 | 68.76 | 71.86 | 3.14 | 198.40 | 100.77 | 3.94 |
| Mean of tropical | 6.94 | 67.70 | 68.70 | 0.99 | 248.32 | 121.00 | 5.07 | 3.45 | Mean of tropical | 1.90 | 68.98 | 72.10 | 3.28 | 195.05 | 98.59 | 3.56 |
| Means of checks | 7.00 | 67.58 | 69.59 | 2.01 | 255.47 | 125.23 | 4.35 | 2.73 | Mean of checks | 2.09 | 68.64 | 73.39 | 5.04 | 202.38 | 103.93 | 5.46 |
| Best check (UH5354) | 9.03 | 70.2 | 71.6 | 1.38 | 267.98 | 130.9 | 2.11 | 1.96 | Best check (UH5354) | 2.93 | 69.68 | 75.50 | 5.80 | 210.88 | 105.69 | 0.71 |
| Grand Mean | 7.43 | 67.50 | 68.38 | 0.88 | 254.88 | 123.16 | 5.71 | 3.03 | Grand Mean | 1.89 | 68.82 | 71.97 | 3.23 | 197.58 | 100.26 | 3.87 |
| LSD(0.05) | 0.95 | 1.40 | 1.69 | 0.99 | 12.27 | 10.14 | 3.40 | 0.69 | LSD(0.05) | 1.09 | 2.62 | 4.04 | 3.19 | 27.72 | 18.45 | 9.11 |

LSD – the least significant difference; TI – temperate introgression.

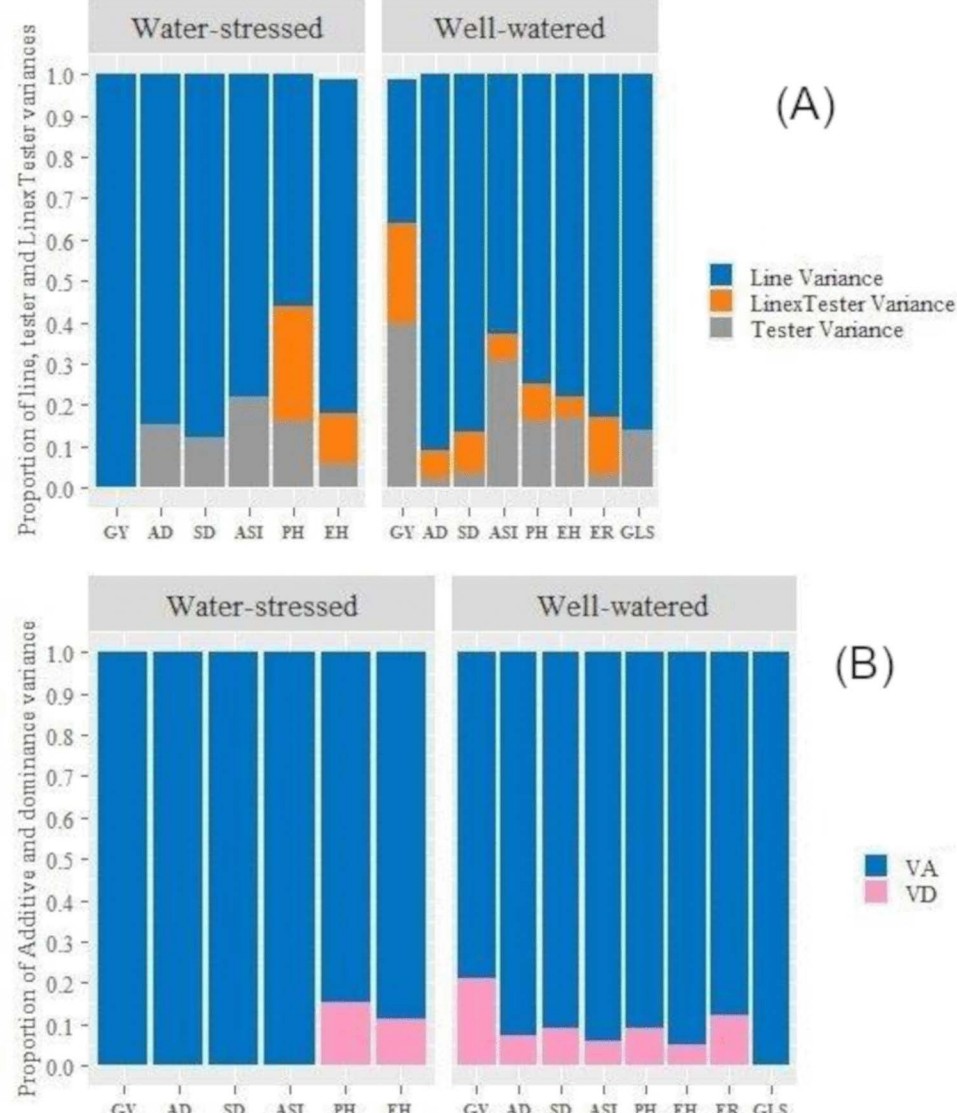

**Fig 3. Genetic variance components of testcross hybrids under contrasting moisture conditions. (A)** Proportional contribution of lines, testers, and line × tester interaction to total genetic variance for grain yield (GY), days to anthesis (AD), days to silking (SD), anthesis–silking interval (ASI), plant height (PH), ear height (EH), ear rot (ER), and gray leaf spot (GLS) under well-watered (WW) and drought stress (DS) conditions. **(B)** Proportion of additive (VA) and dominance (VD) genetic variance for grain yield and agronomic traits of 168 testcross hybrids under WW and DS conditions.

Baker's ratio values were high under WW (0.97–0.99), pointing to the predominance of additive gene action, but declined under DS (0.76–0.97), indicating greater influence of non-additive effects in stressful environments Table 5.

### 3.4 General and specific combining ability

GCA estimates identified several promising parental lines. Lines such as L2, L10, and L16 consistently showed positive GCA effects for GY under both WW and DS conditions, and contributed to favorable flowering traits, including reduced ASI. Conversely, lines such as L17, L18, and L19 had negative GCA effects for yield, limiting their usefulness as parents.

**Table 4. Line by tester analysis of variance of crosses for grain yield (GY), days to anthesis (AD) and silking (SD), anthesis silking interval (ASI), plant height (PH) and ear height (EH), ear rot (ER) and gray leaf spot (GLS) disease under well-watered and water-stressed conditions environments in Kenya.**

| Source of variation | df | GY | df | AD | df | PH | EH | df | ASI | SD | df | ER | df | GLS |
|---|---|---|---|---|---|---|---|---|---|---|---|---|---|---|
| **Well-watered** | | | | | | | | | | | | | | |
| Site | 5 | 219.12*** | 5 | 9608.26*** | 4 | 215423*** | 102756.6*** | 5 | 250.2*** | 6295.86*** | 5 | 6251.7*** | 1 | 162.34*** |
| Rep(Site) | 6 | 2.59* | 6 | 6.75** | 5 | 3117.6*** | 1396.3*** | 6 | 2.3 | 7.38* | 6 | 145.65*** | 2 | 3.66*** |
| Genotypes | 167 | 6.86*** | 167 | 30.2*** | 167 | 1229.1*** | 1022.5*** | 167 | 9.73*** | 35.64*** | 167 | 55.5*** | 167 | 0.97*** |
| Line | 20 | 22.91*** | 20 | 219.83*** | 20 | 6964.6*** | 6262.2*** | 20 | 47.2*** | 245.27*** | 20 | 288.67*** | 20 | 5.38*** |
| Tester | 7 | 48.93*** | 7 | 17.37*** | 7 | 3917.4*** | 3658*** | 7 | 57.71*** | 33.04*** | 7 | 75.73*** | 7 | 2.83*** |
| Line:Tester | 140 | 2.47*** | 140 | 3.75*** | 140 | 275.5** | 142.3*** | 140 | 1.97* | 5.81*** | 140 | 21.18* | 140 | 0.24 |
| Site:Genotypes | 835 | 1.67*** | 835 | 3.29*** | 668 | 197.3 | 131.7*** | 835 | 1.72 | 4.7*** | 835 | 29.67*** | 167 | 0.53*** |
| Site:Line | 100 | 5.97*** | 100 | 13.75*** | 80 | 491.3*** | 399.3*** | 100 | 3.94*** | 19.4*** | 100 | 103.6*** | 20 | 2.14*** |
| Site:Tester | 35 | 1.87** | 35 | 2.89* | 28 | 334.4* | 314.6*** | 35 | 1.78 | 5.45*** | 35 | 58.19*** | 7 | 1.33*** |
| Site:Line:Tester | 700 | 1.05 | 700 | 1.82 | 560 | 148.4 | 84.3 | 700 | 1.4 | 2.56 | 700 | 17.68 | 140 | 0.26 |
| Residuals | 891 | 1.11 | 894 | 1.77 | 745 | 196.8 | 98.3 | 894 | 1.56 | 2.86 | 892 | 17.33 | 298 | 0.26 |
| Baker's ratio | | 0.98 | | 0.99 | | 0.98 | 0.99 | | 0.99 | 0.98 | | 0.97 | | 0.98 |
| **Water-stressed** | | | | | | | | | | | | | | |
| Site | 1 | 83.06*** | 1 | 4158.23*** | 1 | 42.88 | 39.62 | 1 | 1168.67*** | 11439.81*** | 1 | 32.88 | . | . |
| Rep(Site) | 2 | 2.02 | 2 | 23.75** | 2 | 3014.76** | 389.55 | 2 | 66.69*** | 7.46 | 2 | 162.41* | . | . |
| Genotypes | 167 | 0.71 | 167 | 7.05*** | 167 | 533.02* | 262.54*** | 167 | 5.51 | 13.03*** | 167 | 52.48 | . | . |
| Line | 20 | 1.41* | 20 | 33.11*** | 20 | 1469.96*** | 991.08*** | 20 | 7.52 | 40.67*** | 20 | 52.98 | . | . |
| Tester | 7 | 0.62 | 7 | 14.8*** | 7 | 647.49 | 467.54** | 7 | 5.06 | 12.47 | 7 | 34.74 | . | . |
| Line:Tester | 140 | 0.62 | 140 | 2.94 | 140 | 393.51 | 148.22 | 140 | 5.25 | 9.12 | 140 | 53.32 | . | . |
| Site:Genotypes | 167 | 0.84 | 167 | 4.85** | 167 | 400.15 | 174.89 | 167 | 6.76 | 12.62*** | 167 | 64.87* | . | . |
| Site:Line | 20 | 2.3*** | 20 | 18.68*** | 20 | 1066.51*** | 406.33*** | 20 | 13.04* | 40.43*** | 20 | 137.86*** | . | . |
| Site:Tester | 7 | 0.56 | 7 | 7.5* | 7 | 224.78 | 359.87* | 7 | 16.17* | 26.38** | 7 | 18.75 | . | . |
| Site:Line:Tester | 140 | 0.65 | 140 | 2.74 | 140 | 313.73 | 132.58 | 140 | 5.39 | 7.96 | 140 | 56.75 | . | . |
| Residuals | 269 | 0.78 | 298 | 3.37 | 298 | 408.38 | 142.27 | 292 | 6.09 | 8.30 | 295 | 48.96 | . | . |
| Baker's ratio | | 0.86 | | 0.97 | | 0.91 | 0.95 | | 0.82 | 0.92 | | 0.76 | | |

*, **, and *** significant at p = 0.05, p = 0.01 and p = 0.001, respectively, df – degrees of freedom.

Among testers, T1, T2, T3, T4 and T7 were strong general combiners, while T5, T6 and T8 contributed negatively under both conditions (Table 4). Nine lines had positive GCA for GY under WW, and five lines under DS conditions. L2, L7 and L6 were the top combiners for GY under optimal conditions, while L18, L16, L14, L10 and L2 were the best combiners for GY under DS conditions (Table 4). Among the testers, T1, T2, T3, T4 and T7 had positive GCA for GY under WW and DS conditions, whereas as T5, T6 and T8 showed undesirable negative effects in both environments.

The GCA estimates helped to identify good combiners for various traits. For earliness six ex-PVP temperate introgressed lines (L8, L9, L10, L14, L15 and L16), two tropical lines (L20 and L21) and one SC tester (T5) that showed negative GCA for AD and SD under both WW and DS conditions (Table 4). For ASI, seven temperate introgressed lines (L6, L7, L9, L10, L14, L15 and L16), three tropical lines L1, L18 and L19 and two SC testers (T5 and T8) had desirable negative GCA. For GLS, where negative GCA is also valuable, eight lines (L8, L9, L10, L12, L14, L15, L1 and L21) and four testers (T2, T3, T4 and T7) had negative GCA indicating beneficial for GLS resistance.

SCA analysis identified hybrid combinations with significantly higher yields than checks under WW, while differences under DS, were similar, though some hybrids displayed favorable SCA effects (Fig 4). Under DS, SCA ranged from −0.79

**Table 5. Estimates of the general combining ability (GCA) of temperate introgressed, tropical lines and testers for grain yield (GY), ear rot (ER), days to anthesis (AD), days to silking (SD), anthesis silking interval (ASI), plant height (PH), ear height (EH) and gray leaf spot (GLS) under optimal (WW) and drought stress (DS) conditions.**

| Line/Tester | GY WW | GY DS | AD WW | AD DS | ASI WW | ASI DS | SD WW | SD DS | PH WW | PH DS | EH WW | EH DS | ER WW | GLS WW |
|---|---|---|---|---|---|---|---|---|---|---|---|---|---|---|
| L1 | 0.07 | 0.00 | 0.12 | 0.26 | −0.87*** | −1.02* | −0.76 | −0.65 | −11.86*** | −1.26 | 5.51* | 2.93 | 0.91 | −0.05 |
| L2 | 0.60* | 0.01 | −1.33*** | −0.20 | 0.95*** | 0.78 | −0.36 | 0.70 | 4.96 | 1.26 | 4.73 | 4.64 | 0.46 | 0.13 |
| L3 | −0.05 | 0.00 | −0.62 | −0.56 | 0.65** | 0.78 | 0.05 | 0.08 | −0.55 | −1.12 | 9.76*** | 2.69 | −0.51 | 0.24 |
| L4 | −0.03 | −0.01 | 1.74*** | 0.44 | 0.31 | 2.05*** | 2.01 | 4.22** | 6.49* | 0.31 | 8.84*** | 1.44 | −1.57 | 0.25 |
| L5 | −0.11 | 0.00 | 0.80 | 0.33 | 0.27 | 0.40 | 1.05 | 1.43* | −11.19** | −1.43 | 3.34 | −1.19 | −1.86* | 0.02 |
| L6 | 0.43 | 0.00 | 0.13 | 0.19 | −0.34 | −0.77 | −0.21 | −1.05 | 6.24* | 0.49 | −4.00 | −2.18 | 1.27 | 0.20 |
| L7 | 0.55* | −0.01 | 0.73 | 0.56 | −1.01*** | 0.36 | −0.30 | 1.35 | 7.74** | −1.61 | −5.91 | −4.79 | −0.05 | 0.06 |
| L8 | 0.18 | 0.00 | −1.25** | −0.15 | 0.65** | −0.01 | −0.57 | −1.29 | 3.47 | −0.75 | −11.63*** | −4.29 | 2.46** | −0.07 |
| L9 | 0.27 | 0.00 | −1.23** | −0.22 | −1.15*** | −0.11 | −2.38 | −1.12 | −15.55*** | −2.22 | −10.52*** | −4.46 | 0.93 | −0.04 |
| L10 | 0.28 | 0.01 | −1.23 | −0.46 | −0.43 | −0.41 | −1.61 | −2.28** | −2.08 | −0.19 | −14.34*** | −3.60 | 1.05 | −0.06 |
| L11 | −0.03 | −0.01 | 2.55*** | 0.70 | 0.28 | −0.10 | 2.79 | 2.38** | 5.67 | −1.13 | 15.59*** | −0.19 | 0.27 | −0.07 |
| L12 | 0.16 | −0.01 | 2.52*** | 0.65 | −0.21 | −0.35 | 2.25 | 1.85* | 4.63 | 2.47 | 9.99** | 5.35 | −0.40 | −0.14 |
| L13 | −0.04 | 0.00 | −1.04 | −0.41 | 0.09 | 0.93 | −0.93 | 0.53 | 1.78 | 0.14 | −3.94 | −0.07 | −0.86 | 0.15 |
| L14 | −0.09 | 0.01 | −1.46 | −0.21 | −0.06 | −0.69 | −1.49 | −2.35** | 5.37 | 0.29 | 3.78 | 1.38 | 1.00 | −0.58** |
| L15 | −0.09 | 0.00 | −0.86* | −0.29 | −0.14 | −0.75 | −0.99 | −1.78* | 13.70*** | 3.29 | 9.69** | 4.68 | 0.42 | −0.58** |
| L16 | 0.01 | 0.01 | −0.66 | −0.64 | −0.95*** | −1.17* | −1.61 | −2.95** | 6.05* | 3.57 | −0.01 | 5.02 | −0.42 | −0.27 |
| L17 | −0.55* | −0.01 | 0.76 | −0.18 | 0.96*** | 0.37 | 1.72 | −0.54 | 11.47*** | 0.29 | 1.45 | −1.25 | −0.18 | 0.06 |
| L18 | −0.86** | 0.02 | 1.88*** | 0.55 | −0.16 | −0.04 | 1.69 | 2.30** | −10.37*** | 0.76 | −3.51 | 0.09 | −0.27 | 0.21 |
| L19 | −0.40 | 0.00 | 1.56*** | 0.51 | −0.17 | −0.40 | 1.37 | 0.68 | −8.60** | −0.62 | −0.80 | −0.67 | −1.71* | 0.39 |
| L20 | −0.16 | 0.00 | −0.93* | −0.18 | 0.53*** | −0.30 | −0.36 | −1.06 | −4.16 | −0.42 | −7.61** | −1.94 | 0.30 | 0.21 |
| L21 | −0.16 | 0.00 | −2.19*** | −0.68 | 0.79*** | 0.47 | −1.36 | −0.44 | −13.20*** | −2.12 | −10.41*** | −3.59 | −1.23 | −0.08 |
| T1 | 0.21 | 0.02 | 0.17 | 0.01 | −0.06 | 0.22 | 0.12 | 0.46 | −0.98 | −0.88 | −0.17 | −0.31 | 0.20 | 0.01 |
| T2 | 0.45** | 0.06 | 0.14 | 0.20 | −0.24 | 0.34 | −0.04 | 0.78 | 5.40** | 0.52 | 0.53 | 0.24 | −0.16 | −0.07 |
| T3 | 0.07 | 0.03 | 0.10 | 0.21 | −0.25 | 0.13 | −0.09 | 0.37 | −0.19 | 0.41 | 0.06 | 0.41 | 0.10 | −0.15 |
| T4 | 0.20 | 0.03 | −0.17 | −0.13 | 0.33 | −0.33 | 0.08 | −0.67 | 2.13 | 0.53 | −0.14 | −0.07 | 0.01 | −0.01 |
| T5 | −0.05 | −0.01 | −0.23 | −0.21 | −0.30 | −0.40 | −0.46 | −0.80 | −0.59 | 1.02 | 3.13 | 0.81 | −0.01 | 0.11 |
| T6 | −0.95 | −0.15 | −0.30* | −0.29 | 0.93 | 0.62 | 0.41 | −0.06 | −3.22 | −0.01 | −8.23*** | −1.00 | 0.07 | 0.14 |
| T7 | 0.24 | 0.03 | 0.09 | 0.00 | 0.17 | −0.37 | 0.23 | −0.60 | 4.18* | 0.81 | 5.20** | 0.54 | −0.13 | −0.02 |
| T8 | −0.16 | −0.01 | 0.19 | 0.22 | −0.57 | −0.20 | −0.25 | 0.51 | −6.73*** | −2.41 | −0.38 | −0.60 | −0.08 | 0.00 |

*, **, and *** significant at p = 0.05, p = 0.01 and p = 0.001, respectively.

for T2/L6 to 1.07 t/ha for T5/L14. Under WW value ranged from 1.90 t/ha for T6/L17 to 1.20 t/ha for T6/L19 (Fig 4). The hybrids T5/L14, T5/L1 and T8/L20 were ranked as the best specific combiners with highest SCA for GY under DS while other hybrids T6/L19, T6/L6 and T6/L18 had the highest SCA for GY under WW (Fig 4). Six hybrids (T1/L11, T3/L3, T5/L14, T5/L1, T8/L20, T7/9) with positive SCA estimates under both environments.

## 4. Discussion

Continued selection among maize populations tend to fix specific alleles at genes controlling GY and secondary traits, resulting in reduced genetic diversity in the local genetic pool [35]. Introduction of exotic germplasm is one of the strategies employed by maize breeding institutions like the CIMMYT to enhance local genetic variability [36,37]. In this study,

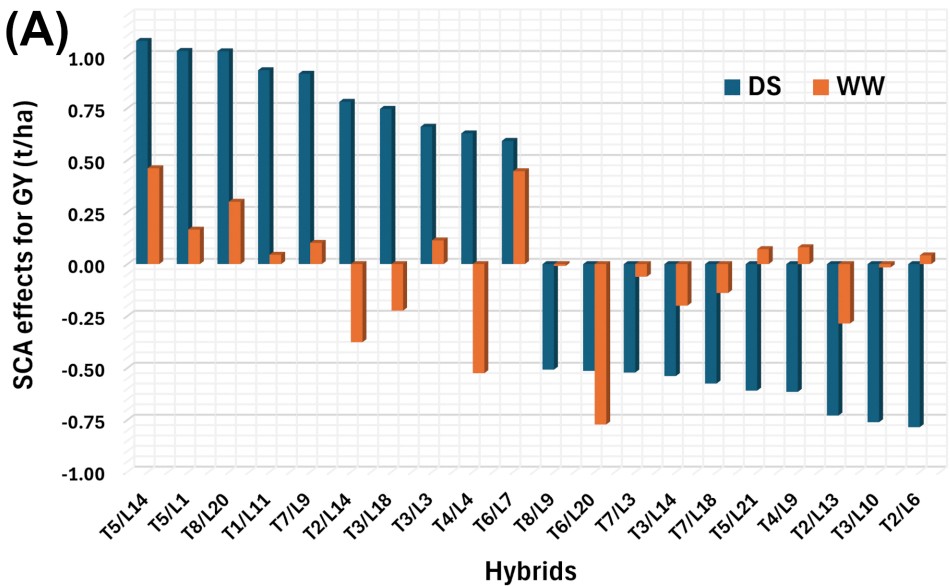

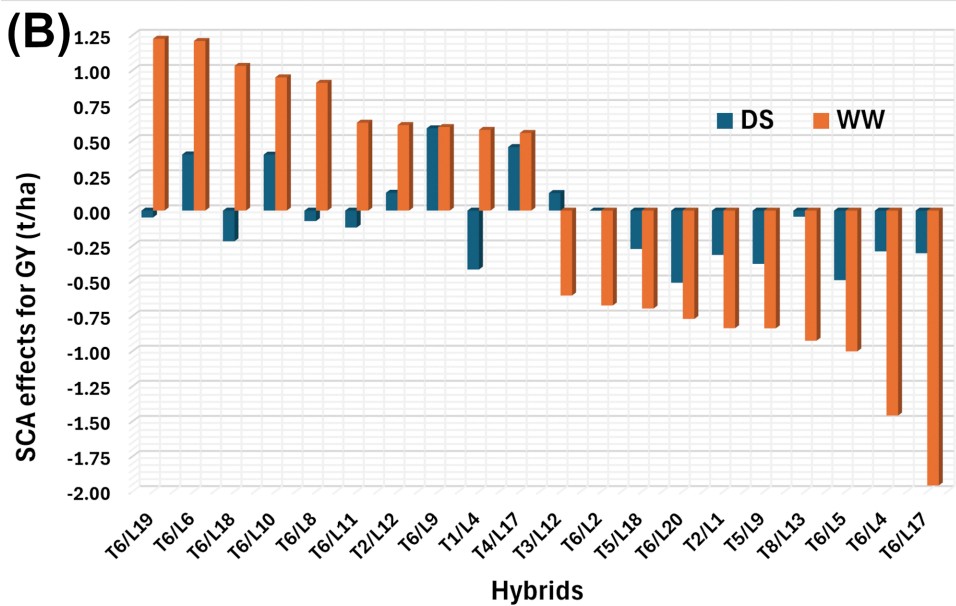

**Fig 4. Cross-environment comparison of SCA effects for grain yield (GY).** (A) The ten hybrids with the highest and lowest SCA effects identified under drought stress (DS) and their corresponding SCA effects under well-watered (WW) conditions. (B) The ten hybrids with the highest and lowest SCA effects identified under WW and their corresponding SCA effects under DS conditions.

temperate ex-PVP inbred lines were introduced, backcrossed to elite tropical adapted lines and their testcross hybrids were evaluated using line×tester analysis to identify promising lines to increase yield potential of tropical adapted lines and hybrids for further evaluation and commercialization in SSA.

This study revealed significant genetic variation in maize testcross hybrids for GY and related traits under WW and DS conditions. Compared with WW management, DS reduced the proportion of genotypic variance and increased genotype×environment interactions for several traits, reflecting complex adaptive responses to stress. This observation

corroborates findings by Ertiro et al. and Manigben et al. [38,39] who reported reduced genotypic variance for GY and agronomic traits under induced DS condition. Consequently, heritability estimates were consistently higher under WW than DS conditions, suggesting that selection is more reliable in favorable environments. However, the strong impact of DS on yield, flowering synchrony, and plant stature highlights the need for targeted evaluation under stress-prone environments to identify drought-tolerant genotypes.

Under DS, residual and environmental variation exceeded genetic variance for yield (Table 3), lowering heritability not because of weak genetic control but because drought amplified environmental noise and masked genotypic differences [38,39]. As a result, phenotypic performance became a poorer predictor of breeding value, increasing the likelihood of advancing environmentally favored entries and discarding truly tolerant genotypes, and causing unstable rankings across locations. Therefore, effective DS screening should focus on improving precision through multi-location testing, adequate replication, spatial adjustment, and BLUP-based estimation across environments, complemented by secondary traits with higher repeatability such as anthesis–silking interval, stay-green, and ears per plant. Together, these approaches help recover genetic signal and enable reliable selection despite inherently low heritability under drought conditions.

The present study revealed significant genetic variation among tropical and temperate introgressed maize testcross hybrids for GY and related agronomic traits under WW and DS conditions. Hybrids showed up to 74% reduction in GY, 22% lower PH, 18% lower EH, and a 72% increase in ASI under DS compared to WW (Table 3). Comparison of the reduction in PH and EH under DS between temperate introgressed and original tropical hybrids revealed that the magnitude of reduction was comparable, and occasionally greater, in the tropical backgrounds. This indicates that the shorter stature was largely a consequence of drought-induced growth limitation rather than the expression of a compact plant type derived from the ex-PVP introgressions. The clear reduction in yield and agronomic performance under DS is consistent with earlier findings that drought is one of the most severe constraints to maize production in SSA, often leading to large yield losses [23,40].

The effect of DS on GY and the performance of growth traits was also reported by several studies [18,20,41,42]. Under DS, reduced GY could be attributed to delayed pollination caused by an elongated ASI [43,44], that result in reduced number of kernels per ear and subsequent reduction in weight of kernels. Overall, the results reveal a wide yield gap between WW and DS conditions, reflecting the severity of DS. Nevertheless, certain hybrids demonstrated strong potential for advancement: T2/L7, T3/L2, and T5/L2 emerged as high yielding under WW, while T5/L1, T8/L20, and T8/L2 performed best under DS (Table 3). The comparable yield of T2/L7 with the best commercial check under WW conditions indicates that yield potential was maintained while improving key adaptive traits such as flowering synchrony and ear placement (Table 3). The clear superiority of hybrid T5/L1 under DS, driven by reduced ASI and absence of ear rot, suggests enhanced reproductive stability and successful grain set under moisture limitation. These hybrids represent promising candidates for deployment in diverse production environments, combining yield potential under favorable conditions with resilience under stress.

Temperate introgression improved GY in most tropical maize backgrounds, with hybrids developed from TI- of CML495, CML539, CML312, and CML491 outperforming their tropical counterparts by 0.6–1.5 t/ha under WW condition (Fig 2B). The largest gains were observed in CML539 and CML495 TI derivatives, where TI_1 and TI_2 consistently produced the highest yields, highlighting the complementarity of favorable alleles from temperate and tropical lines. These findings support the previous research that introgression can enhance yield potential and stress resilience in tropical maize [18,45]. Conversely, CML537 showed minimum difference between tropical and TI hybrids, indicating background dependent benefits of introgression. Overall, TI proves to be a promising strategy for expanding the genetic base and improves yield in tropical germplasm.

## 4.1 Line by tester analysis

Significant genotypic and site variances for GY and related traits under WW conditions, and for some traits under DS, highlight the role of genetics and environments on trait expression. These results confirm that phenotypic variation in

testcross hybrids is influenced by both parental combinations and testing environments, underscoring the importance of multi-location testing in hybrid breeding. Developing reliable prenatal variability is key to maize improvement [46,47]. In this study, partitioning genotypic variance into lines, testers, and their interactions provided valuable insights into genetic architecture underlying these traits.

GCA from lines and testers, and SCA from line × tester interactions, were significant for GY and most traits under WW, and for several traits under DS, indicating the importance of both additive and non-additive gene actions, consistent with earlier studies [42,48]. Additive variance had a greater impact than dominance variance for GY and secondary traits, suggesting that favorable alleles with cumulative effects are crucial for predictable selection gains. High Baker's ratios under WW supported the predominance of additive gene action. The slight decline in Baker's ratio under DS reflects a greater influence of environmental variation and non-additive effects under stress conditions. Nevertheless, the consistently high values (>0.82) across traits indicate that additive gene action remained predominant. Thus, drought tolerance in this germplasm is largely governed by additive effects, with non-additive interactions contributing to performance but not controlling inheritance. Similar trends have been reported in drought breeding studies where additive effects were key for synchrony in flowering and yield stability [20,44,45].

Several lines exhibited desirable GCA effects for GY and flowering traits across environments. Notably, temperate introgressed lines L2 and L16 showed consistent positive GCA under both WW and DS. Lines L6, L7, and L12 also exhibited strong additive effects and reduced ASI, making them sources of drought adaptation alleles. In this study, the testers were intentionally selected from complementary heterotic groups to maximize heterosis with the OFP donor lines. Most of testers used here are from Heterotic group B and lines are from HG-A (Table 1). Among testers, T2, T3, T4, and T7 had positive GCA for GY across environments, in agreement with earlier work [18,42]. These testers are already part of elite CIMMYT maize breeding pipelines in eastern and southern Africa and continue to provide valuable information for parental selection. Conversely, some lines (L17, L18, L19) displayed negative GCA effects for GY, limiting their direct use as parents

SCA effects revealed valuable hybrid combinations with potential for releases and commercialization after further multilocation trials to ascertain the stability and yield under different management conditions. Under WW conditions, hybrids like T6/L6, T6/L10, T2/L12, T6/L9, and T4/L17 displayed positive SCA for GY, while under DS, promising combinations included T5/L15, T5/L1, T8/L20, T1/L11, T7/L9, and T3/L3. Consistent positive SCA across environments suggests that resilient hybrids can be developed by strategically combining temperate introgressed and tropical lines. Similar findings have been reported in other tropical maize breeding programs [20,49]. Notably, high-SCA hybrids paired with strong GCA parents demonstrate the potential of exploiting both additive and non-additive effects improved breeding outcomes.

The significant GEIs observed highlights the complexity of breeding for drought tolerance. Under stress, GEIs explained a larger portion of the variance indicating performance instability and underscoring the need of testing across diverse environments. This aligns with earlier studies showing that multi-environment evaluation is crucial for identifying hybrids that combine stability with high yield [19,50]. The combination of results from variance partitioning, GCA, SCA, and hybrid performance highlights the need for balanced strategies: genomic selection effectively captures additive effects and accelerate favorable allele accumulation, while exploiting heterosis through hybrid development remains critical for maximizing yield under both optimal and stress conditions [51,52].

## 5. Conclusion

Temperate ex-PVP introgression provides useful genetic variation for improving tropical maize across contrasting environments. In most backgrounds, introgression increased GY while maintaining adaptation under both WW and DS conditions. The predominance of additive effects indicates strong potential for population improvement through recycling schemes, particularly rapid-cycle recurrent selection coupled with genomic selection to accumulate favorable alleles. At the same time, significant SCA effects support continued exploitation of heterosis in hybrid development. Overall, these results

clarify the genetic architecture governing performance under optimal and stress conditions and demonstrate that strategic integration of temperate alleles can broaden the genetic base without compromising adaptation. This approach can accelerate development of high-yielding, climate-resilient maize hybrids for sub-Saharan Africa and other drought-prone regions.

## Supporting information

**S1 Table. Environmental characterization of the growing season, including rainfall and temperature, across six well-watered (WW) and two drought-stress (DS) experimental sites.**
(XLSX)

**S2 Table. The mean performance of temperate introgressed and tropical lines derived testcross hybrids across locations evaluated under optimum conditions.**
(XLSX)

**S3 Table. The mean performance of temperate introgressed and tropical lines derived testcross hybrids evaluated under managed drought stress conditions.**
(XLSX)

## Acknowledgments

The authors thank CIMMYT scientists and the technical team across various country offices and stations in Kenya, Mexico and Zimbabwe for their support during the line development process. We extend our gratitude to the management of the Kenya Agricultural and Livestock Research Organization (KALRO) stations for allowing access to experimental facilities across various locations in Kenya where field experiments conducted and KALRO staff for aiding data collection were conducted.

## Author contributions

**Conceptualization:** Yoseph Beyene.

**Data curation:** Isaiah Aleri, Andrew Chavangi.

**Formal analysis:** Isaiah Aleri, Juan Burgueno.

**Funding acquisition:** Manje Gowda, Yoseph Beyene.

**Investigation:** Manje Gowda, Andrew Chavangi, Yoseph Beyene.

**Methodology:** Yoseph Beyene.

**Software:** Isaiah Aleri, Juan Burgueno.

**Supervision:** Yoseph Beyene.

**Validation:** Andrew Chavangi, Yoseph Beyene.

**Visualization:** Isaiah Aleri, Manje Gowda.

**Writing – original draft:** Isaiah Aleri, Manje Gowda.

**Writing – review & editing:** Isaiah Aleri, Manje Gowda, Andrew Chavangi, Juan Burgueno, Yoseph Beyene.

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
