## [Decision Letter · Decision Letter 0]

26 Dec 2025

Dear Dr. Gowda,

Thank you for submitting your manuscript to PLOS ONE. After careful consideration, we feel that it has merit but does not fully meet PLOS ONE’s publication criteria as it currently stands. Therefore, we invite you to submit a revised version of the manuscript that addresses the points raised during the review process.

We look forward to receiving your revised manuscript.

Kind regards,

Mehdi Rahimi, Ph.D.

Academic Editor

PLOS One

Journal Requirements:

“The research was supported by the Bill and Melinda Gates Foundation (B&MGF), and the United States Agency for International Development (USAID) through the Stress Tolerant Maize for Africa (STMA, B&MGF Grant # OPP1134248) Project, AGGMW (Accelerating Genetic Gains in Maize and Wheat for Improved Livelihoods, B&MGF Investment ID INV-003439) project. “

Reviewers' comments:

Reviewer's Responses to Questions

**Comments to the Author**

1. Is the manuscript technically sound, and do the data support the conclusions?

Reviewer #1: Yes

Reviewer #2: Yes

2. Has the statistical analysis been performed appropriately and rigorously?

Reviewer #1: Yes

Reviewer #2: Yes

3. Have the authors made all data underlying the findings in their manuscript fully available?

Reviewer #1: Yes

Reviewer #2: Yes

4. Is the manuscript presented in an intelligible fashion and written in standard English?

Reviewer #1: Yes

Reviewer #2: Yes

Reviewer #1: Introduction

• The introduction discusses ex-PVP lines broadly. You must explain why these specific ex-PVP lines (OFP donors) were chosen for introgression. Were they selected for specific traits (stay-green, root architecture) or yield potential?

• Explicitly state the gap this study fills. Many studies look at ex-PVP in favorable conditions; emphasize the scarcity of data regarding their performance under managed drought stress in the tropics.

Materials and Methods

• Section 2.1 mentions "Visual selection... focused on germination, stand establishment...". Please quantify the selection intensity. How many families were started vs. how many were selected?

• In Equation 1 and 2, you treat Lines and Testers as random effects. Since you selected specific diverse lines and elite testers, a fixed-effect model might be arguable for the GCA estimates, though random is acceptable for variance components. Briefly justify the choice of random effects for the parents.

• Provide a supplementary figure or table summarizing rainfall and temperature at the 6 WW and 2 DS sites during the growing season to validate the environmental distinctness.

Results

• Figure 1 (histograms) and Figure 4 (SCA charts) are blurry and difficult to read. The axes labels are illegible. These must be re-exported at 300+ DPI.

• Table 2, the error variance is quite high under DS compared to genetic variance. Discuss the implications of this low signal-to-noise ratio on the reliability of the DS heritability estimates.

• You compare Top 10% hybrids to the "Mean of checks". It is more rigorous to compare the best experimental hybrid against the best commercial check, not the average of all checks.

Discussion

• The results show reduced Plant Height (PH) and Ear Height (EH) under drought. Discuss if the ex-PVP introgressions contributed to a "compact plant type" that might be beneficial for Harvest Index under stress, or if the reduction was purely stress-induced stunting.

• You observed a drop in Baker's ratio under drought. Expand on the biological reason. Does this suggest that drought tolerance in this germplasm is driven more by specific dominance/epistatic interactions (survivability) than by additive yield potential?

• Discuss why specific testers (e.g., T2, T3) were good general combiners. Do they belong to a heterotic group that complements the "OFP" donors? This links back to the missing pedigree data in the Intro.

• Provide a concrete breeding strategy. Should breeders use these TI lines for recycling (recurrence selection) given the high additive variance in WW, or solely for hybrid constitution (SCA) in target stress environments?

• Explicitly contrast your results with other ex-PVP studies (e.g., Nelson et al., 2008 or Mikel et al.). Do your findings regarding yield penalties/gains align with global experiences of bringing temperate maize into the tropics?

Reviewer #2: This research work is insightful, well executed and scientific. The authors have excellent writing skills. However, the researchers need to justify if the research is still relevant/novel considering the period it started (2008).

**Do you want your identity to be public for this peer review?** For information about this choice, including consent withdrawal, please see our Privacy Policy

Reviewer #1: No

Reviewer #2: No

---

## [Author Response · Author response to Decision Letter 1]

11 Feb 2026

PONE-D-25-60749

Combining Ability and Yield Performance of Temperate ex-PVP Introgressed and Tropical Maize Lines under Contrasting Moisture Regimes

PLOS One

Dear Dr. Gowda,

Thank you for submitting your manuscript to PLOS ONE. After careful consideration, we feel that it has merit but does not fully meet PLOS ONE’s publication criteria as it currently stands. Therefore, we invite you to submit a revised version of the manuscript that addresses the points raised during the review process.

• A letter that responds to each point raised by the academic editor and reviewer(s). You should upload this letter as a separate file labeled 'Response to Reviewers'.

We look forward to receiving your revised manuscript.

Kind regards,

Mehdi Rahimi, Ph.D.

Academic Editor

PLOS One

Response: Thanks for the comment. We formatted the revised manuscript as suggested.

Review Comments to the Author

Reviewer #1: Introduction

Comment 1

• The introduction discusses ex-PVP lines broadly. You must explain why these specific ex-PVP lines (OFP donors) were chosen for introgression. Were they selected for specific traits (stay-green, root architecture) or yield potential?

Response: Thank you for the comment. The ex-PVP lines (OFP donors) were not chosen arbitrarily but through a multi-step selection process. After assembling an initial set of ex-PVP germplasm, we consulted maize breeding experts from USDA and U.S. seed companies to identify lines with demonstrated breeding value. Selection primarily emphasized high yield potential, along with key plant architecture traits (e.g., narrow, erect leaves suitable for high plant density and mechanized production), stay-green expression, foliar disease tolerance, plant vigor, and standability. The candidate lines were further evaluated in preliminary line trials to verify their agronomic performance. Based on this combined expert knowledge and phenotypic validation, the most suitable ex-PVP lines were selected for introgression into the tropical breeding pipeline.

Comment 2

• Explicitly state the gap this study fills. Many studies look at ex-PVP in favorable conditions; emphasize the scarcity of data regarding their performance under managed drought stress in the tropics.

Response: Thank you for this important suggestion. We agree that the key knowledge gap lies in the limited information on the performance and breeding value of ex-PVP germplasm under managed drought stress in tropical environments. Most previous studies have evaluated ex-PVP lines primarily under favorable or high-input conditions in temperate regions, with little evidence on their utility within tropical drought-breeding pipelines.

In this study, we specifically address this gap by evaluating temperate ex-PVP lines in combination with drought-tolerant tropical elite germplasm under managed drought, MLN, and optimum conditions. Tropical elite lines contribute adaptive and stress-tolerance alleles, whereas ex-PVP lines are a valuable source of high-yield potential and well-characterized genomic regions associated with productivity. Because many yield-related haplotypes in ex-PVP germplasm are already defined, their introgression can be tracked and validated in tropical genetic backgrounds.

Therefore, our work provides evidence on (i) the combining ability of ex-PVP germplasm under managed tropical drought stress and (ii) their potential to simultaneously improve yield potential and stress resilience. This study demonstrates the feasibility of using temperate introgression as a strategy to develop high-yielding, multi-stress-tolerant tropical maize germplasm and hybrids. We included this information in the revised manuscript, P3, L85-105.

Comment 2

Materials and Methods

• Section 2.1 mentions "Visual selection... focused on germination, stand establishment...". Please quantify the selection intensity. How many families were started vs. how many were selected?

Response: Thanks for your comment. The BC₁F₂ generation was initiated with 250 families per cross. In subsequent generations, visual selection based on germination, stand establishment, plant architecture, ear placement, ear filling, kernel quality, and resistance to major foliar diseases [gray leaf spot (Cercospora zeae-maydis), northern leaf blight (Exserohilum turcicum), and maize streak virus (Mastrevirus, Geminiviridae)] was applied, and approximately 20% of plants were advanced to the next generation. (P4, L115-119)

Comment 3

• In Equation 1 and 2, you treat Lines and Testers as random effects. Since you selected specific diverse lines and elite testers, a fixed-effect model might be arguable for the GCA estimates, though random is acceptable for variance components. Briefly justify the choice of random effects for the parents.

Response: Thank you for the insightful comment. In Equation 1, hybrids were considered as genotypes to estimate overall genotypic performance across environments. Because a large number of hybrids were evaluated over multiple locations, we fitted a mixed linear model in which genotypes were treated as fixed effects to obtain BLUEs for hybrid performance, while the genotype term was also modeled as random to estimate variance components and BLUPs for each genotype.

In Equation 2, the total genetic variance was partitioned into general combining ability (GCA; lines and testers) and specific combining ability (SCA; line × tester interaction). Lines and testers were modeled as random effects because they represent a sample drawn from broader breeding populations rather than the only parents of inference. Treating them as random effects allows estimation of GCA and SCA variance components. This approach follows the mixed-model framework recommended by Piepho et al. (Piepho H. P., Büchse A., and Emrich K. (2003). A hitchhiker's guide to mixed models for randomized experiments. J. Agron. Crop Sci. 205 (6), 669–681. doi: 10.1046/j.1439-037X.2003.00049.x) for estimating variance components in multi-environment breeding trials.

Comment 4

• Provide a supplementary figure or table summarizing rainfall and temperature at the 6 WW and 2 DS sites during the growing season to validate the environmental distinctness.

Response: Thanks for your comment. We included the environmental information in the revised manuscript in P5, L165-168.

Comment 5

Results

• Figure 1 (histograms) and Figure 4 (SCA charts) are blurry and difficult to read. The axes labels are illegible. These must be re-exported at 300+ DPI.

Response: Thanks for your comment. We improved the figures and included them in the revised version

Comment 6

• Table 2, the error variance is quite high under DS compared to genetic variance. Discuss the implications of this low signal-to-noise ratio on the reliability of the DS heritability estimates.

Response: Thanks for your comment. Yes, we agree with your comment. The residual errors are relatively higher under DS experiment compared to well-watered trials. We included the relevant information in the revised manuscript. In brief “Under managed DS, residual, environmental, and genotype × environment variation exceeded genetic variance for yield, resulting in low heritability (Table 2). This does not imply weak genetic control but rather that drought amplifies environmental noise, masking true genotypic differences. Consequently, phenotypic performance becomes a poor predictor of breeding value, increasing the risk of advancing environmentally favored entries while discarding genuinely drought-tolerant genotypes. Therefore, the objective of DS screening should be to enhance precision rather than avoid stress environments. Reliable selection requires multi-location testing, adequate replication, spatial correction, and BLUP-based estimates across environments, complemented by secondary traits with higher repeatability such as anthesis–silking interval, stay-green, and ears per plant. Together, these approaches recover the underlying genetic signal and enable effective selection despite inherently low heritability under drought conditions.” P16, L398-408

Comment 7

• You compare Top 10% hybrids to the "Mean of checks". It is more rigorous to compare the best experimental hybrid against the best commercial check, not the average of all checks.

Response: Thank you for the valuable suggestion. We agree that comparing the best experimental hybrid directly with the best commercial check provides a more stringent and practically relevant benchmark. While comparing the top 10% hybrids with the mean of checks helps assess overall population improvement relative to current standards, the best hybrid vs. best check comparison reflects varietal replacement potential. Accordingly, the revised manuscript now includes results and discussion comparing the best-performing hybrid with the best commercial check P10, L297-300; 310-312; P17, L522-527 and Table 3.

Comment 8

Discussion

• The results show reduced Plant Height (PH) and Ear Height (EH) under drought. Discuss if the ex-PVP introgressions contributed to a "compact plant type" that might be beneficial for Harvest Index under stress, or if the reduction was purely stress-induced stunting.

Response: Thank you for the comment. We compared the reduction in plant height (PH) and ear height (EH) under drought stress between the temperate introgressed (TI) hybrids and the original tropical hybrids. The magnitude of reduction was similar, and in some cases greater, in the tropical/original hybrids, indicating that the decrease in PH and EH primarily resulted from drought-induced growth limitation rather than a compact plant type contributed by the ex-PVP introgressions. This clarification has been incorporated into the revised Discussion P17, L508-513.

Comment 9

• You observed a drop in Baker's ratio under drought. Expand on the biological reason. Does this suggest that drought tolerance in this germplasm is driven more by specific dominance/epistatic interactions (survivability) than by additive yield potential?

Response: Thank you for the comment. The observed reduction in Baker’s ratio for grain yield and related traits under drought stress (DS) compared with well-watered conditions (Table 4) likely reflects the increased contribution of non-additive and environmental sources of variation under stress, including residual variance and genotype × environment interaction. However, the Baker’s ratio remained high (>0.82) across traits, including grain yield, indicating that additive genetic effects still predominated. Therefore, drought tolerance in this germplasm appears to be governed mainly by additive effects, while non-additive interactions contribute to performance under stress but do not dominate its inheritance. We included this information in the revised manuscript, P18, L554-559.

Comment 10

• Discuss why specific testers (e.g., T2, T3) were good general combiners. Do they belong to a heterotic group that complements the "OFP" donors? This links back to the missing pedigree data in the Intro.

Response: Thank you for the comment. The testers were intentionally selected from complementary heterotic groups to maximize heterosis with the OFP donor lines. Most of testers used here are from Heterotic group B and all lines are from HG-A. Testers such as T2 and T3 belong to widely used heterotic backgrounds in eastern and southern Africa breeding programs and therefore combined well with the introgressed germplasm, resulting in strong general combining ability. The pedigree information for all testers is clearly indicated in Table 1 (T1–T6) in the revised manuscript, P18, L565-570.

Comment 11

• Provide a concrete breeding strategy. Should breeders use these TI lines for recycling (recurrence selection) given the high additive variance in WW, or solely for hybrid constitution (SCA) in target stress environments?

Response: Thank you for the comment. Given the predominance of additive gene action under both well-watered and drought stress conditions, the TI lines are suitable not only for hybrid constitution but also for population improvement. They can be effectively used in recycling schemes, particularly rapid-cycle recurrent selection combined with genomic selection, to accumulate favorable alleles for grain yield and other key traits. This breeding strategy has been clarified in the revised Discussion section, P19, L600-604.

Comment 12

• Explicitly contrast your results with other ex-PVP studies (e.g., Nelson et al., 2008 or Mikel et al.). Do your findings regarding yield penalties/gains align with global experiences of bringing temperate maize into the tropics?

Response: Thank you for this important suggestion. Previous ex-PVP studies such as Nelson et al. (2008) and Mikel and Dudley (2006) primarily characterized ex-PVP germplasm using molecular markers and aligned them with heterotic groups, demonstrating their usefulness for broadening the genetic base of elite germplasm. More recent studies have moved beyond characterization and used ex-PVP lines as sources of favorable alleles for traits such as earliness, grain yield, and Striga resistance in tropical breeding programs (e.g., Guo et al., 2021; Maazou et al., 2022).

Globally, introgression of temperate maize into tropical germplasm has often resulted in initial adaptation penalties, particularly reduced yield under stress environments due to photoperiod sensitivity and poor stress tolerance. However, after selection and recombination, temperate alleles frequently contribute beneficial effects including improved yield potential, standability, and agronomic performance. Our findings are consistent with this general pattern. In the present study, several temperate-introgressed testcrosses showed competitive or superior grain yield under optimum conditions and, importantly, a subset maintained stable performance under drought and disease stress environments. This indicates that the yield penalty typically associated with temperate germplasm can be minimized when only favorable genomic regions are incorporated into adapted tropical backgrounds.

Therefore, unlike early reports where temperate germplasm mainly caused adaptation drag, our results support the more recent global experience: strategic introgression of selected temperate genomic segments enhances yield potential without compromising adaptation, particularly when combined with

---

## [Decision Letter · Decision Letter 1]

17 Feb 2026

Yield and Combining Ability of Temperate ex-PVP Introgressed and Tropical Maize Lines under Contrasting Moisture Regimes

PONE-D-25-60749R1

Dear Dr. Gowda,

We’re pleased to inform you that your manuscript has been judged scientifically suitable for publication and will be formally accepted for publication once it meets all outstanding technical requirements.

Kind regards,

Prasanta K. Subudhi, Ph.D.

Academic Editor

PLOS One

Additional Editor Comments (optional):

The Authors made necessary revisions to improve the manuscript. The previous reviewer is satisfied with the revisions made in the manuscript. Therefore, the manuscript may now be accepted for publication.

Reviewers' comments:

Reviewer's Responses to Questions

**Comments to the Author**

Reviewer #1: All comments have been addressed

2. Is the manuscript technically sound, and do the data support the conclusions?

Reviewer #1: Yes

3. Has the statistical analysis been performed appropriately and rigorously?

Reviewer #1: Yes

4. Have the authors made all data underlying the findings in their manuscript fully available?

Reviewer #1: Yes

5. Is the manuscript presented in an intelligible fashion and written in standard English?

Reviewer #1: Yes

Reviewer #1: The authors have submitted a comprehensive revision that effectively addresses the concerns raised in the previous round of review. The manuscript has been significantly improved in terms of clarity, scientific rigor, and the contextualization of the findings. The responses provided in the rebuttal letter are well-reasoned and are accurately reflected in the revised text.

**Do you want your identity to be public for this peer review?** For information about this choice, including consent withdrawal, please see our Privacy Policy

Reviewer #1: No

---

## [Editor Report · Acceptance letter]

PONE-D-25-60749R1

PLOS One

Dear Dr. Gowda,

I'm pleased to inform you that your manuscript has been deemed suitable for publication in PLOS One. Congratulations! Your manuscript is now being handed over to our production team.

Kind regards,

on behalf of

Dr. Prasanta K. Subudhi

Academic Editor

PLOS One